

# Event generation and density estimation
# with surjective normalizing flows

**Rob Verheyen**

Department of Physics and Astronomy University College London,
Gower St., Bloomsbury, London WC1E 6BT, UK

r.verheyen@ucl.ac.uk

## Abstract

Normalizing flows are a class of generative models that enable exact likelihood evaluation. While these models have already found various applications in particle physics, normalizing flows are not flexible enough to model many of the peripheral features of collision events. Using the framework of [1], we introduce several surjective and stochastic transform layers to a baseline normalizing flow to improve modelling of permutation symmetry, varying dimensionality and discrete features, which are all commonly encountered in particle physics events. We assess their efficacy in the context of the generation of a matrix element-level process, and in the context of anomaly detection in detector-level LHC events.



# 1 Introduction

First-principle Monte Carlo event generators are a fundamental component of most LHC physics analyses. As the LHC enters its third run, and with the high luminosity upgrade in the near future, the amount of available experimental data is set to increase rapidly. To match the resulting statistical precision, the event generators must follow suit. The nature of perturbative calculations in quantum field theory is that such an increase in precision of the simulations comes hand-in-hand with an increase in complexity, and thus with more costly simulations. This means that advances in event generator technology are required to maintain interpretability of future LHC data [2,3].

One promising avenue to tackle these computational challenges can be found in modern machine learning techniques. In particular, generative models such as generative adversarial networks (GANs) [4], variational autoencoders (VAEs) [5] and normalizing flows [6,7] offer alternatives for fast event generation. Of these models, normalizing flows have the particular advantage of simultaneously enabling event generation and likelihood evaluation, the latter of which is useful in other applications. As a result, they have been successfully used for a variety of tasks including event generation [8–20], anomaly detection [21–23], unfolding [24], the calculation of loop integrals [25], and likelihood-free inference [26–28].

Normalizing flows make use of a set of differentiable bijective functions to transform between a simple, fixed base distribution and a complex, learned distribution. Much progress has been made in the development of normalizing flow architectures that are both expressive and efficient [29–37]. However, normalizing flows lack in flexibility due to the bijective nature of the transforms, meaning that they are essentially restricted to modelling a continuous feature space of fixed dimension. For the purposes of event generation and likelihood estimation in particle physics, more flexibility is often required, for instance to model discrete features or varying dimensionality on an event-by-event basis.

On the other hand, GANs and VAEs do not have these limitations, but they do not offer exact likelihood evaluation. GANs are trained adversarially and thus do not offer access to the likelihood at all, but VAEs are able to provide a lower bound estimate of the likelihood. In [1] a method was outlined that combines the favorable properties of normalizing flows with the flexibility of VAEs. In this work, we explore the use of this framework, as well as other solutions, to improve the flexibility of normalizing flows in the context of particle physics event generation and density estimation.

In section 2, we summarize normalizing flow and VAE architectures, as well as their combination as detailed in [1]. Section 3 describes a test case and a baseline normalizing flow, which

are then used to explore the incorporation of permutation invariance (section 3.3), varying dimensionality (section 3.4) and discrete features (section 3.5). In section 4 these techniques are then applied to a density estimation problem in the context of the Dark Machines Anomaly Score Challenge [38]. We conclude in section 5.

## 2 Surjective Normalizing Flows

We are interested in setting up a generative model that is able to generate new events, but also evaluate the likelihood of existing events. *Latent variable models* are one such class of models. They are typically composed of a set of relatively simple components, but turn out to be expressive enough to learn the complicated probability distributions that are commonly encountered in particle physics. Given a set of physical events $x \in \mathcal{X}$ of dimension $d_x$, we define an auxiliary set of latent variables $z \in \mathcal{Z}$ of dimension $d_z$ with an associated joint probability distribution $p(x, z)$, which is specified by the model and thus depends on a set of trainable parameters. The marginal probability density

$$p(x) = \int_{\mathcal{Z}} dz\, p(x, z) = \int_{\mathcal{Z}} dz\, p(z)\, p(x|z) \tag{1}$$

is then the distribution of interest. The second equality in eq. (1) arises through the general product rule, and implies a generative process, which is given by

$$\begin{aligned} z &\sim p(z), \\ x &\sim p(x|z). \end{aligned} \tag{2}$$

This generative process is efficient as long as $p(z)$ and $p(x|z)$ are simple enough to be sampled from, while the conditioning on the latent variables $z$ leads to increased expressivity. However, the trade-off is that the evaluation of the marginal likelihood, eq. (1), is generally not tractable. Aside from the fact that likelihood evaluation is an objective in its own right, the training of probabilistic models is generally accomplished by maximum likelihood estimation, or equivalently, minimization of the Kullback-Leibler divergence $\mathbb{D}_{\mathrm{KL}}$ with an empirical distribution $p_{\mathrm{data}}(x)$, i.e.

$$\begin{aligned} \mathcal{L}_{\mathrm{MLE}} &= \mathbb{E}_{p_{\mathrm{data}}(x)}\big[-\log p(x)\big] \\ &= \mathbb{D}_{\mathrm{KL}}\big[p_{\mathrm{data}}(x)|p(x)\big] - \underbrace{\int_{\mathcal{X}} dx\, p_{\mathrm{data}}(x) \log p_{\mathrm{data}}(x)}_{\text{constant}}, \end{aligned} \tag{3}$$

where $\mathbb{E}_{p_{\mathrm{data}}(x)}$ indicates an expectation value over $p_{\mathrm{data}}(x)$. Eq. (3) also requires likelihood evaluation, and as such the intractability of eq (2) is a significant issue, for which several solutions exist.

### 2.1 Normalizing Flows

One option to resolve the intractability of eq. (1) is to remove the stochastic component from the conditional probability distribution, $p(x|z) = \delta(x - f(z))$, leading to (in log-space)

$$\log p(x) = \log\left[\int_{\mathcal{Z}} dz\, p(z)\, \delta(x - f(z))\right] = \log p(z) + \log|J(x)|, \tag{4}$$

where $|J(x)|$ is the Jacobian determinant associated with the transform $f(z)$. The Dirac delta function requires $d_x = d_z$, and the evaluation of $p(z) = p(f^{-1}(x))$ is only possible if $f(z)$ is a

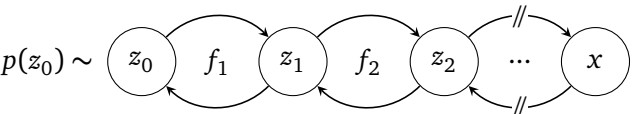

Figure 1: Visualization of a normalizing flow architecture. The forward direction starts with a sample $z_0$ distributed according to the base distribution $p(z_0)$ after which $n$ flow bijections $f_i$ are applied to arrive at $z_n = x$. In the inverse direction, starting from $x$, the flow transforms are applied in reverse order to arrive at $z_0$, for which $p(z_0)$ can be evaluated.

bijective function. Eq. (4) is the fundamental step in *normalizing flow* architectures. Normalizing flow transforms are composable, meaning that to further improve expressivity, multiple may be stacked, $z_0 \to z_1 \to ... \to z_n = x$, leading to

$$\log p(x) = \log p(z_0) + \sum_{i=1} \log |J_i(z_i)|. \tag{5}$$

The base distribution $p(z)$ can then be taken to be simple, such as a multivariate uniform or a standard normal with diagonal covariance. The generative process then involves drawing a sample from $p(z)$, which is then passed through the layers of flow transforms in the *forward* direction until reaching $x$. On the other hand, likelihood evaluation starts from $x$ which is passed in the *inverse* direction until $z$ is reached while aggregating the Jacobian determinant of every transform along the way. The prior can then be evaluated and eq. (5) can be computed. Figure 1 shows an illustration of the normalizing flow architecture.

Much of the research on normalizing flows has focussed on improving the expressiveness and efficiency of the bijective transform, see e.g. [39, 40] for reviews. We describe the specific architecture used in this work in section 3.

## 2.2 Variational Inference

Instead of solving the intractability of eq. (1) by constraining $p(x|z)$ to a delta function, another option is *variational inference*[1]. In that case, one introduces a variational approximation $q(z|x)$ to the true posterior $p(z|x)$. The log-likelihood may then be rewritten as

$$
\begin{aligned}
\log p(x) &= \int_{\mathcal{Z}} dz\, q(z|x) \log \frac{p(x|z)p(z)}{p(z|x)} \\
&= \int_{\mathcal{Z}} dz\, q(z|x) \left[ \log p(x|z) - \log \frac{q(z|x)}{p(z)} + \log \frac{q(z|x)}{p(z|x)} \right] \\
&= \mathbb{E}_{q(z|x)} \left[ \log p(x|z) \right] - \mathbb{D}_{\mathrm{KL}} \left[ q(z|x), p(z) \right] + \mathbb{D}_{\mathrm{KL}} \left[ q(z|x), p(z|x) \right],
\end{aligned}
\tag{6}
$$

where in the first line we have added a factor $\int_{\mathcal{Z}} dz\, q(z|x) = 1$ since $p(x)$ does not depend on $z$, and then used Bayes rule to rewrite $p(x)$. The intractability of $\log p(x)$ is now isolated in the third term of eq. (6), which is strictly positive. The combination of the other two terms is commonly referred to as the evidence lower bound (ELBO). Due to the positivity of the third term, the ELBO can be optimized in place of the full likelihood. Eq. (6) serves as the foundation of the VAE, in which $p(x|z)$ and $q(z|x)$ are parameterized by deep neural networks and during training the ELBO is evaluated with a single Monte Carlo sample $z \sim q(z|x)$. Contrary to the normalizing flow approach, variational autoencoders do not require any restrictions on the

---

[1]The objective of variational inference is often stated as the computation of the posterior $p(z|x) = p(z, x)/p(x)$, for which the marginalized distribution $p(x)$ is also required.

form of $p(x|z)$. However, the gap between the likelihood and the ELBO vanishes only in the limit where $q(z|x) = p(z|x)$, which in practice is difficult to accomplish.

## 2.3 Surjective and Stochastic Transforms

In [1] it was pointed out that the normalizing flow and VAE paradigms can the unified by rewriting eq. (6) as

$$\log p(x) = \mathbb{E}_{q(z|x)}\left[\log p(z) + \underbrace{\log \frac{p(x|z)}{q(z|x)}}_{\mathcal{V}(x,z)} + \underbrace{\log \frac{q(z|x)}{p(z|x)}}_{\mathcal{E}(x,z)}\right], \tag{7}$$

where $\mathcal{V}(x,z)$ is the *likelihood contribution* and $\mathcal{E}(x,z)$ is the *bound looseness*. For a normalizing flow transform, no variational approximation of the posterior is required, i.e. $p(x|z) = \delta(x - f(z))$ and $q(z|x) = \delta(z - f^{-1}(x))$. The result is that $\mathcal{V}(x,z) = \log|J(x)|$ and $\mathcal{E}(x,z) = 0$, recovering eq. (4). However, for stochastic transforms like that of the VAE, $\mathcal{V}(x,z)$ may be evaluated with a single Monte Carlo sample, while $\mathcal{E}(x,z)$ remains intractable, again serving as a (strictly positive) error on the full likelihood.

Furthermore, it is possible to define *surjective* transforms, which are deterministic in one direction and stochastic in the other. In case of a surjection in the inverse direction $x \to z$, i.e. $q(z|x) = \delta(z - g(x))$ but $p(x|z)$ remains stochastic, the bound looseness vanishes if $p(x|z)$ only has support over the set $B(z) = \{x|z = g(x)\}$.[2] In case of a surjection in the forward direction $z \to x$ however, $p(x|z) = \delta(x - h(z))$ and $q(z|x)$ stochastic, the bound looseness is nonzero.

Section 3 will explore several of these transforms, as they will turn out to be useful in the modelling of several features commonly encountered in particle collision events. Note that eq. (7) naturally supports the composable nature of a normalizing flow akin to eq. (5), such that bijective, surjective and stochastic transforms may be combined.

## 3 Application in Particle Physics Events

In this section we explore the use of surjective transforms as part of a normalizing flow to improve the handling of several distinctive features of particle physics events: permutation invariance, varying dimensionalities and discrete features. We first describe a relatively low-dimensional benchmark process which displays all of these features, and determine a baseline normalizing flow architecture that is used throughout, before continuing with a description of techniques and an assessment of their efficacy.

### 3.1 A Benchmark Process

We consider the matrix element-level process

$$gg \to \tilde{g}\tilde{g}\tilde{g}\tilde{g}, \tag{8}$$

at 3 TeV, using the default parameters of the `MSSM_SLHA` model of `Madgraph5_aMC@NLO` [41], which sets the gluino mass to $m_{\tilde{g}} = 607.71$ GeV. A few example Feynman diagrams are shown in figure 2.

This process presents a four-fold permutation symmetry in the final state, allowing us to explore techniques that can incorporate permutation invariance in the generative model.

---

[2]In this case, the posterior $p(z|x) = \delta(z - g(x))$ because any value of $x$ can only have originated from $z = g(x)$. As a result, $q(z|x) = p(z|x)$ and $\mathcal{E}(x,z) = 0$.

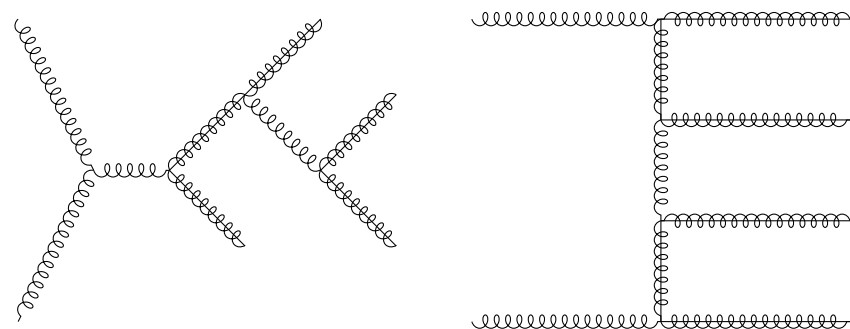

Figure 2: Example Feynman diagrams that contribute to the $gg \rightarrow \tilde{g}\tilde{g}\tilde{g}\tilde{g}$ matrix element.

Furthermore, the phase space is eight-dimensional, which conveniently divides into four sets of two variables for every gluino, making permutation of the phase space straightforward. We choose to use the polar and azimuthal angles of the gluinos in the center-of-mass frame as parameterization. Note that this forces the normalizing flow to learn a nontrivial distribution due to phase space alone, as the phase space measure vanishes in some regions.[3] The polar and azimuthal angle are mapped to a space $\mathcal{X} = [0,1]^8$ through

$$x_{\theta,i} = \frac{1}{2}\left(\cos\theta_i + 1\right) \text{ and } x_{\varphi_i,i} = \frac{\varphi}{2\pi} \text{ for } i = 1,...,4 . \tag{9}$$

The process shown in eq. (8) also presents a rich discrete structure, which enables an examination of techniques that model continuous and discrete features simultaneously. Six objects in the adjoint representation of $SU(3)_c$ lead to 120 leading-$N_c$ colour-orderings, and the gluino masses produce a varied spectrum in the 64 helicity configurations.

Finally, for experiments with varying dimensionality, we mix in $gg \rightarrow \tilde{g}\tilde{g}$ events. In total, we generate 1.2M $gg \rightarrow \tilde{g}\tilde{g}\tilde{g}\tilde{g}$ and 120k $gg \rightarrow \tilde{g}\tilde{g}$ events, reserving 100k and 10k for both validation and testing respectively. The two-gluino events are parameterized by their common polar and azimuthal angles in the center-of-mass frame.

## 3.2 Baseline Normalizing Flow

We employ a baseline normalizing flow architecture to learn continuous densities throughout the following experiments. We choose to make use of an autoregressive flow [32] similar to the one used in [10, 21]. In this model, the bijection $f$ on a $d$-dimensional event space $\mathcal{X}$ is factorized into a set of $d$ one-dimensional transforms characterized by

$$x_j = f_j(z_j; \theta_j(z_{1:j-1})), \tag{10}$$

where for $j \in [1,d]$, $z_j$ is the $j$th component of $z$. The bijection $f_j$ is thus parameterized by a function $\theta_j$ of the preceding components $z_0$ through $z_{j-1}$. As such, the forward transform from $z$ to $x$ must be performed sequentially starting from $z_0$. On the contrary, the inverse transform from $x$ to $z$ can be performed in parallel. This choice means that training and inference is fast, but sampling is relatively slow [4]. In some of the following experiments, the sampling step of a normalizing flow is instead required during training. In such cases, the architecture is inverted such that that direction is fast. In our implementation, the functional form of $f_j$ is given by a rational quadratic spline [37] and $\theta_j$ is a MADE network [42]. These spline transforms are easily constrained to a finite domain, making them well-suited for density estimation in particle physics as phase space can usually be mapped to a finite volume.

---

[3]For instance, it is not possible for all gluino momenta to lie in the same hemisphere.

[4]Sampling events on a GPU is still fast, taking approximately 20 seconds for $10^6$ events in our experiments.

Table 1: Table of hyperparameters and training setup used in the experiments of section 3.

| Model | | Training | |
|---|---|---|---|
| Parameter | Value | Parameter | Value |
| RQS knots | 32 | Batch size | 25k |
| MADE layers | 2 | Optimizer | Adam |
| MADE units per dim | 10 | Learning rate | $10^{-3}$ |
| Flow layers | 8 | Validation interval | 25 |
| | | LR decay | 0.5 |
| | | LR decay patience | 50 |

In several cases, conditioning of the normalizing flow on some discrete value is required. That is, instead of just modelling a density $p(x)$ over the continuous space $\mathcal{X}$, the flow needs to represent a density $p(x|y)$, where $y \in \mathcal{Y}$ is a discrete number. This type of conditioning proceeds through learnable embeddings of the values of $y$ into a continuous space of the size of the hidden layers of the MADE network. These embeddings are then added before the first activation of the MADE network of every flow layer.

The normalizing flow and all extensions to it discussed in this section are implemented in PYTORCH [43]. The code is publically available[5]. The hyperparameters of the flow are listed in table 1. The base distribution $p(z_0)$ is chosen to be a uniform distribution over $[0, 1]^8$, such that the flow transforms are constrained to $[0, 1] \rightarrow [0, 1]$. Models are trained with the Adam optimizer [44] with default values of $\beta_1$ and $\beta_2$. Because some of our experiments feature different amounts of training data, we formulate the training procedure, of which the parameters are also listed in table 1[6], in terms of iterations rather than epochs. After fixed intervals, the model is validated against the validation set. If the loss has not improved for a fixed number of validations (the decay patience), the learning rate is multiplied by the decay factor. This procedure repeats until the learning rate drops low enough for training to have effectively ceased (in practice, a factor of $10^{-3}$ of the initial learning rate), or until 5000 validations have occurred. The model is then finally evaluated on the independent test set.

We emphasize that the experiments performed in this work are not focussed on obtaining the best possible performance of the baseline normalizing flow. Instead, their point is to explore various techniques that one can use to improve performance on the types of data that are not easily modelled by a regular normalizing flow, but that regularly appear in the context of particle physics. Previous work [11, 45, 46] has explored improving the performance of normalizing flows through the application of an auxiliary classifier neural network, which can in principle be applied in the experiments that follow.

## 3.3 Permutation Invariance

Particle physics events often display a large degree of permutation invariance. In the matrix element-level example used here, the final state has a four-fold permutation symmetry. More generally, jet constituents are permutation invariant, a fact that is already exploited in other ML architectures [47, 48]. Permutation invariance of identified objects also appears at the detector level.

In [1], two methods were proposed to instill permutation invariance into a normalizing flow model: a sorting surjection and a stochastic permutation. The forward and backward

---

[5]https://github.com/rbvh/surflows

[6]We find that large batch sizes lead to better performance. A batch size of 25k requires $\sim$ 5 GB of VRAM, which is readily available on most modern GPUs.

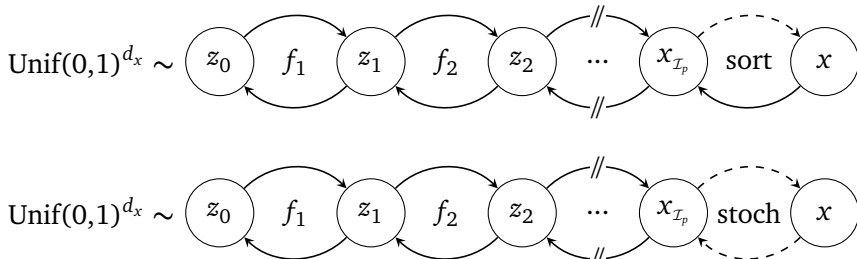

Figure 3: The normalizing flow architecture of figure 1 including a sort surjection or a stochastic permutation transform at the end. Solid arrows indicate deterministic transform directions, while dashed arrows are stochastic. The base distributions have been specified to a multivariate uniform.

transforms are defined as

$$p_{\text{sort}}(x|z) = \sum_{\mathcal{I}_p} \frac{1}{D!} \delta(x - z_{\mathcal{I}_p^{-1}}), \qquad p_{\text{stoch}}(x|z) = \sum_{\mathcal{I}_p} \frac{1}{D!} \delta(x - z_{\mathcal{I}_p^{-1}}),$$

$$q_{\text{sort}}(z|x) = \sum_{\mathcal{I}_p} \delta_{\mathcal{I}_p, \text{argsort}(x)} \delta(z - x_{\mathcal{I}_p}), \qquad q_{\text{stoch}}(z|x) = \sum_{\mathcal{I}_p} \frac{1}{D!} \delta(z - x_{\mathcal{I}_p}),$$

$$\mathcal{V}_{\text{sort}}(x, z) = \log(D!), \qquad\qquad \mathcal{V}_{\text{stoch}}(x, z) = 0, \tag{11}$$

where $\mathcal{I}_p$ is a set of permutation indices for the components of $x$ or $z$, $\mathcal{I}_p^{-1}$ are their inverse and $D$ is the number of permutable classes. That is, in the inverse direction, the sort surjection orders $x$ following some predicate, while the stochastic permutation randomly shuffles $x$. In the forward direction, both transforms randomly shuffle $z$, leading to permutation-invariant samples. The sort transform is surjective in the inverse direction and adheres to the property described in section 2.3 required for $\mathcal{E}_{\text{sort}}(x, y) = 0$. On the other hand, stochastic permutation does not lead to a vanishing bound looseness. Both transforms can lead to improved modelling in different ways. The stochastic permutation may be viewed as effectively increasing the training statistics by a factor $D!$, while the sort surjection can be thought of as folding the space $\mathcal{X}$ into a volume that is a factor $1/D!$ smaller.

### 3.3.1 Experiments

We perform experiments with the default flow model as discussed in the beginning of this section, either without permutation transform, or with a stochastic permutation transform, or a sort surjection appended at the end. In this case, the sort surjection orders gluinos according to their polar angle, as these are features that are directly present in the phase space parameterization. An illustration of this architecture is shown in figure 3.

To illustrate the gain in performance due to the addition of a permutation transform, we perform experiments with a varying size of the training dataset. Figure 4 shows the distributions of the energy spectra of the individual gluinos sampled from models trained on just 50k events. Note that the gluino energy is not one of the variables that is directly present in the parameterization of phase space. This means that the model must learn the relevant correlations between all polar and azimuthal angles to correctly predict the spectrum.

We observe that both permutation transforms, and especially the stochastic permutation, lead to significant improvement in the fidelity of the modeling of the true distribution. At such small training statistics, the effective increase with a factor of $4! = 24$ due to the four-fold permutation symmetry is substantial. Even without permutation transform, the flow mostly

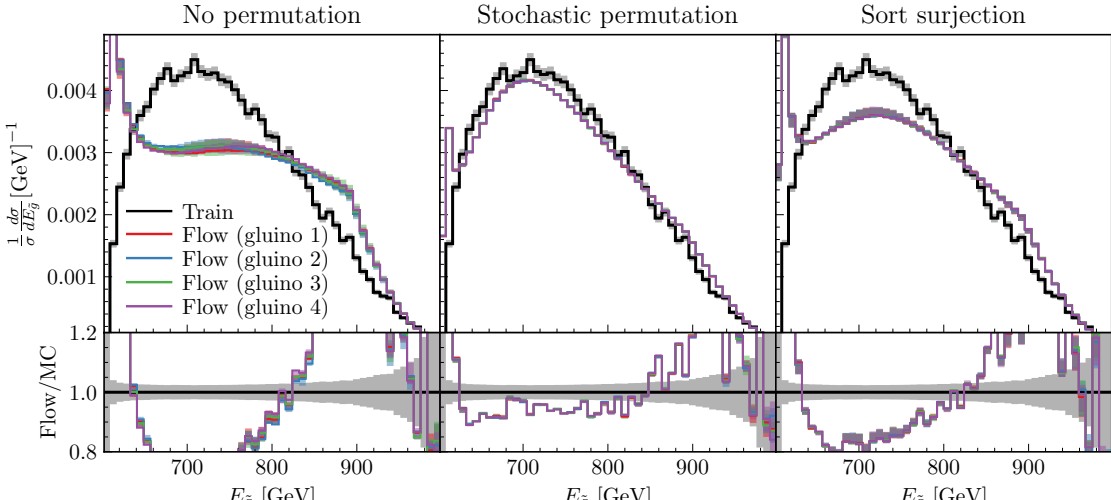

Figure 4: The energy distribution per individual gluino as predicted by models trained on 50k training events without permutation transform (left), with stochastic permutation (middle) or with sort surjection (right). The error bands correspond with variations between three independent runs.

learns to treat the gluinos on equal footing, as only small deviations between the gluino energy spectra appear. On the other hand, both permutation transforms enforce permutation invariance in the generative direction, leading to spectra that are identical up to statistical fluctuations.

Figure 5 instead shows the digluino invariant mass spectrum, but this time models trained on 50k, 200k and 1M events are included. One striking feature of this figure is the fact that the cases without permutation transform and with sort surjection show definite improvement as the size of the training dataset increases. However, the case of the stochastic permutation shows little improvement. This picture is corroborated when one considers the progression of the testing log likelihood as a function of the size of the training dataset, which is shown in figure 6. The model with stochastic permutation significantly outperforms the other models for small training statistics, but it is eventually overtaken, even by the model without permutation transform. This effect occurs due to the nonvanishing bound looseness associated with the stochastic permutation transform. This means that, given unconstrained training data and network capacity, the other two cases will eventually approach the theoretical maximum log likelihood. On the other hand, the model with stochastic permutation is always limited by a nonzero bound looseness, diminishing its performance. We conclude that the inclusion of a permutation transform is always beneficial, but the choice between the two options should be guided be the size of the available training dataset.

## 3.4 Varying Dimensionality

Particle physics events typically do not contain a constant number of objects. As a result, the dimensionality of phase space can vary on an event-by-event basis. Normalizing flow models on the other hand learn probability distributions of fixed dimension. One method of modelling varying dimensionalities was presented in [11] for the specific case of $pp \rightarrow Z_{\mu\mu} + \{1, 2, 3\}$ jets, where conditional flow networks are trained to add jets to baseline $Z_{\mu\mu}$ events. Alternatively, one could train multiple generative models for all individual configurations. The downside of this approach is that the training statistics are split between the models. On the other hand, a single model that is able to generate all configurations will be able to learn any underlying

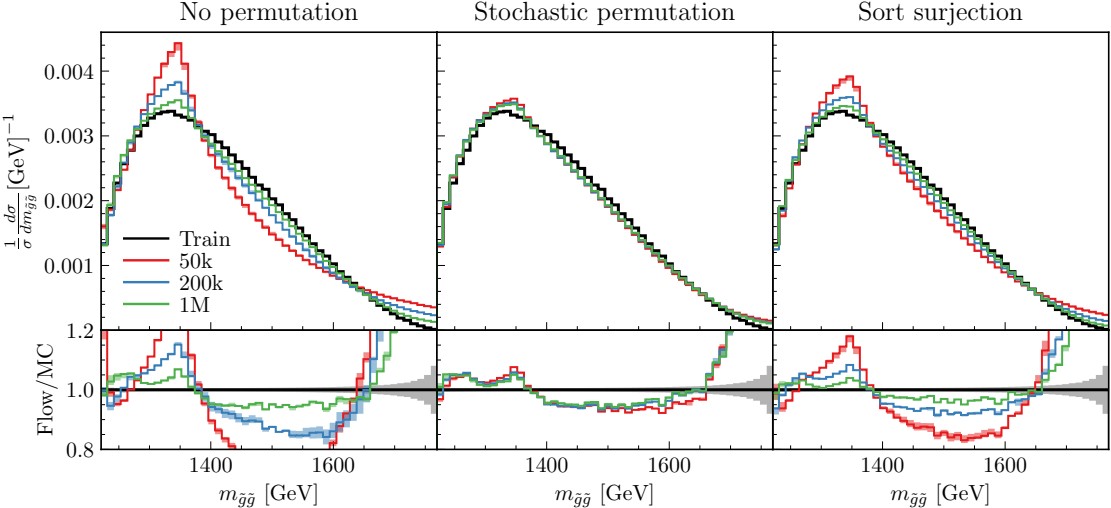

Figure 5: The digluino invariant mass as predicted by models trained on sets of training data of size 50k (red), 200k (blue) and 1M (green), without permutation transform (left), with stochastic permutation (middle) or with sort surjection (right). The error bands correspond with variations between three independent runs.

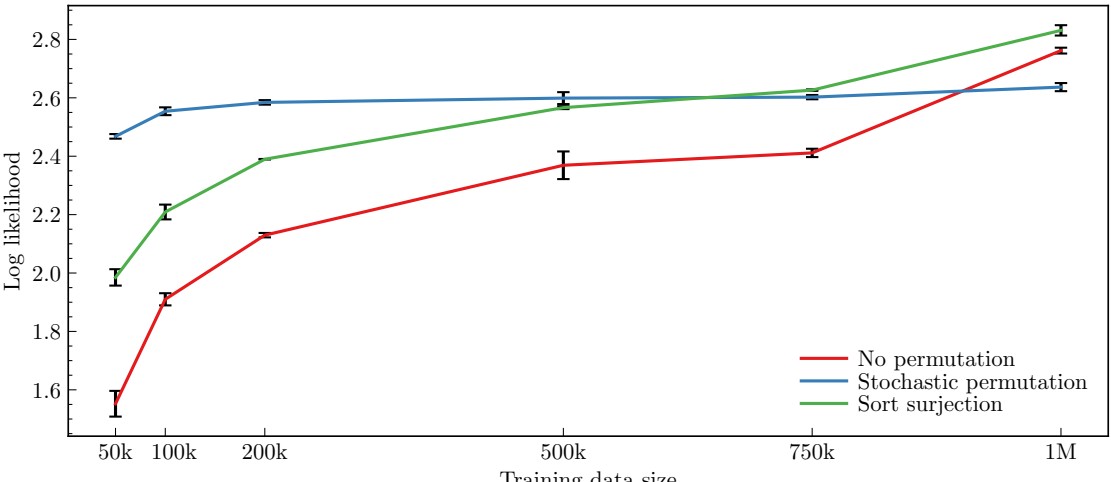

Figure 6: The development of the test log likelihood (higher is better) of models without permutation transform (red), with stochastic permutation (blue) or with sort surjection (green). The points and error bars correspond with the mean and standard deviation over three independent training runs.

patterns that are common between them. The architecture of [11] accomplishes this, but it does not generalize easily to many configurations.

Here, we introduce a surjective transform that is able to combine an arbitrary number of configurations into a single model.[7] We refer to it as a *dropout* transform, as its function is to stochastically drop a subset of the latent variables. To that end, we introduce a set of dropout indices $\mathcal{I}_\downarrow$ for the components of $x$ and $z$, as well as their complement $\mathcal{I}_\uparrow$ such that $\{\mathcal{I}_\downarrow, \mathcal{I}_\uparrow\} = \{1, ..., d\}$. The forward and backward transforms are

$$p_{\text{drop}}(x|z) = \sum_{\mathcal{I}_\downarrow} p_{\mathcal{I}_\downarrow} \, \delta(x_{\mathcal{I}_\uparrow} - z_{\mathcal{I}_\uparrow}),$$

$$q_{\text{drop}}(z|x) = \sum_{\mathcal{I}_\downarrow} \delta_{\mathcal{I}_\downarrow, \text{argdrop}(x)} \, \delta(z_{\mathcal{I}_\uparrow} - x_{\mathcal{I}_\uparrow}) q(z_{\mathcal{I}_\downarrow}). \tag{12}$$

That is, the forward transform picks a set of dropout indices $\mathcal{I}_\downarrow$ with probability $p_{\mathcal{I}_\downarrow}$ and drops the corresponding components from the feature vector $z$. The inverse fills the dropped components with probability $q(z_{\mathcal{I}_\downarrow})$.

The optimal values of the probabilities $p_{\mathcal{I}_\downarrow}$ are the normalized cross-sections of events with configuration $\mathcal{I}_\downarrow$, which can easily be extracted from the training data, i.e. $p_{\mathcal{I}_\downarrow} \equiv (p_{\mathcal{I}_\downarrow})_{\text{data}}$. Note that there is a potential interaction with the permutation transform of the previous section. When a sort surjection is used, every feature-space configuration maps to a single set of dropout indices. However, if a stochastic permutation transform is used, each feature-space configuration can map to multiple sets of dropout indices, and the probabilities $p_{\mathcal{I}_\downarrow}$ should be adjusted accordingly.

In practice, this means that separate samples of $x$ can have different dimensionalities. Any following flow layers expect input of the original dimension of $z$. We handle this by setting dropped indices to values outside the domain of the relevant latent space.[8] Subsequent flow layers are then set up to leave dropped dimensions unchanged. The baseline normalizing flow is conditioned on the dropout indices.

Instead of computing the likelihood contribution and the bound looseness, we can directly evaluate the marginal likelihood as

$$p(x) = \int_{\mathcal{Z}} dz \, p_{\text{drop}}(x|z) p(z)$$

$$= \sum_{\mathcal{I}_\downarrow} p_{\mathcal{I}_\downarrow} \int_{\mathcal{Z}_{\mathcal{I}_\downarrow}} dz_{\mathcal{I}_\downarrow} \, p(x_{\mathcal{I}_\uparrow}, z_{\mathcal{I}_\downarrow}), \text{ where } x_{\mathcal{I}_\uparrow} = z_{\mathcal{I}_\uparrow}. \tag{13}$$

In this expression, the latent variables in $p(z)$ have been separated explicitly into the dropped variables $z_{\mathcal{I}_\downarrow}$ and the remaining ones $z_{\mathcal{I}_\uparrow} = x_{\mathcal{I}_\uparrow}$. In general, the integral in eq. (13) is intractable. However, if the distributions of $z_{\mathcal{I}_\uparrow}$ and $z_{I_\downarrow}$ are independent, i.e. $p(x_{\mathcal{I}_\uparrow}, z_{\mathcal{I}_\downarrow}) = p(x_{\mathcal{I}_\uparrow}) p(z_{I_\downarrow})$, eq. (13) reduces to

$$p(x) = \sum_{\mathcal{I}_\downarrow} p_{\mathcal{I}_\downarrow} p(x_{\mathcal{I}_\uparrow}), \tag{14}$$

which can be evaluated exactly. For independence to hold for all $\mathcal{I}_\downarrow$, all latent variables must be mutually independent. While this is generally not the case after one or more flow layers, the

---

[7]The transform introduced here bears resemblance to the tensor slicing surjection of [1]. However, in this case the sliced dimensions are selected stochastically.

[8]For example, in the experiments performed in this section the latent space is restricted to $[0, 1]$, and dropped indices are set to $-1$.



Figure 7: The dropout architecture described in section 3.4. The permutation transform can either be a stochastic permutation or a sort surjection, in which case the arrow in the inverse direction would be solid.

base distribution is usually chosen as a simple factorized distribution, i.e. a multivariate uniform or normal distribution with diagonal covariance. Thus, a model with tractable likelihood emerges by placing the dropout surjection directly after the base distribution. An illustration of the resulting architecture is shown in figure 7. Note that the stochastic inverse $q(z_{\mathcal{I}_\downarrow})$ is no longer required, since the values of $z_{\mathcal{I}_\downarrow}$ are only used to evaluate the base distribution, and because of the independence from $z_{\mathcal{I}_\uparrow}$ the contribution to the likelihood integrates to unity.

### 3.4.1 Optimization

The objective of eq. (3) then decomposes into

$$
\mathcal{L}_{\text{MLE}} = -\int_{\mathcal{X}} dx \sum_{\mathcal{I}_\downarrow} p_{\mathcal{I}_\downarrow} p_{\text{data}}(x_{\mathcal{I}_\uparrow}) \log\left[ \sum_{\mathcal{I}_\downarrow'} p_{\mathcal{I}_\downarrow'} p(x_{\mathcal{I}_\uparrow'}) \right]
$$

$$
= -\sum_{\mathcal{I}_\downarrow} p_{\mathcal{I}_\downarrow} \int_{\mathcal{X}_{\mathcal{I}_\uparrow}} dx_{\mathcal{I}_\uparrow} p_{\text{data}}(x_{\mathcal{I}_\uparrow}) \log p(x_{\mathcal{I}_\uparrow}) - \underbrace{\sum_{\mathcal{I}_\downarrow} p_{\mathcal{I}_\downarrow} \log p_{\mathcal{I}_\downarrow}}_{\text{constant}}. \tag{15}
$$

That is, maximum likelihood estimation corresponds with a weighted multi-objective optimization [49, 50] of the distributions $p(x_{\mathcal{I}_\downarrow})$. Eq. (15) can then be interpreted as a linear scalarization of such a multi-objective optimization problem with preference vector $p_{\mathcal{I}_\downarrow}$. A more general set of solutions to these problems adhere to the property of Pareto-optimality, which means that one objective cannot be further improved without degrading at least one of the others. Linear scalarizations of multi-objective optimization problems like eq. (15) can be shown to locate a single Pareto-optimal solution, but it is often not clear if this is the preferred one. For instance, when one aims to simultaneously model the distributions of dropout configurations with widely-varying cross-sections, eq. (15) assigns small weight to configurations with small cross-sections, which will lead to poor modelling of the corresponding conditional probability distributions. We can instead opt to select a different preference vector $r_{\mathcal{I}_\downarrow}$, leading to

$$
\mathcal{L}_{\text{MLE}}^{(r)} = -\sum_{\mathcal{I}_\downarrow} r_{\mathcal{I}_\downarrow} \int_{\mathcal{X}_{\mathcal{I}_\uparrow}} dx_{\mathcal{I}_\uparrow} p_{\text{data}}(x_{\mathcal{I}_\uparrow}) \log p(x_{\mathcal{I}_\uparrow}). \tag{16}
$$

### 3.4.2 Experiments

We perform experiments with the model of section 3.3 with ordering surjection, and include a dropout layer directly after the base distribution. The objective is to learn the distributions of two-gluino and four-gluino events simultaneously. There are only two non-vanishing dropout probabilities

$$
\begin{aligned}
p_{\text{two}} &= 4.0069 \cdot 10^{-5}, & \mathcal{I}_\downarrow &= \{3,4,5,6,7,8\}, \ \mathcal{I}_\uparrow = \{1,2\} \\
p_{\text{four}} &= 1 - 4.0069 \cdot 10^{-5}, & \mathcal{I}_\downarrow &= \{\}, \ \mathcal{I}_\uparrow = \{1,2,3,4,5,6,7,8\}.
\end{aligned} \tag{17}
$$

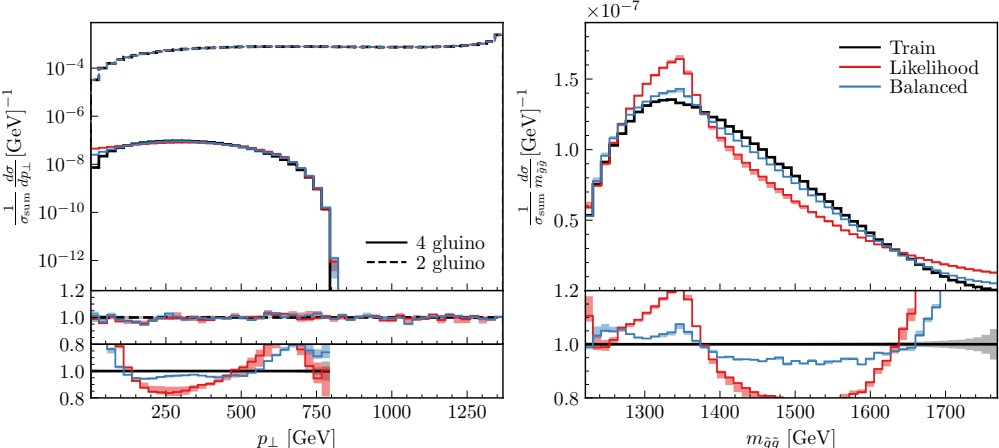

Figure 8: The transverse-momentum spectrum (left) of four-gluino events (solid) and two-gluino events (dashed), as well as the digluino invariant mass spectrum (right) of four-gluino events. The MC truth (black) is compared with the model illustrated in figure 7 trained by maximizing eq. (15) (red) and eq. (16) with the elements of the weight vector $r_{\mathcal{I}_\downarrow}$ set to 1/2 (blue). The error bands correspond with variations between three independent runs.

The values of the dropout likelihoods follow the cross-sections of the corresponding processes. Training is performed by either maximizing eq. (15) (referred to as *likelihood*), or by maximizing eq. (16) (referred to as *balanced*) where the elements of the weight vector are set to 1/2. The large difference in cross-section would lead to a very small amount of four-gluino events in the training data. We instead opt to use the full datasets described in section 3.1 and reweigh as appropriate.

Figure 8 shows the spectra of the transverse momentum of four-gluino and two-gluino events, as well as the digluino invariant mass of the four-gluino events. We observe significantly better performance in the balanced case. This is the result of the small weight assigned to the four-gluino conditional in the likelihood case, resulting in poor optimization. Experiments with different values of the weight vector $r_{\mathcal{I}_\downarrow}$ did not lead to qualitatively different results, which appears to indicate one can expect similar performance as long as none of the conditional distributions of eq. (16) are substantially suppressed.

## 3.5 Discrete Features

Generative models in particle physics have predominantly focussed on modelling the continuous phase space of particle collisions. However, scattering events are often not only characterized by their energy-momentum distributions, but also by a variety of discrete features which are related to the quantum numbers of the particles involved in the scattering.

Several methods of modelling discrete features have been considered in the context of normalizing flows. Some of these find explicit methods of casting eq. (4) in a form that can handle discrete data [51, 52]. However, particle physics presents a distinct situation in which continuous and discrete features are jointly distributed, and these methods are not straightforwardly extended to model correlations between continuous and discrete data components. We thus explore methods that either map the discrete space to a continuous one, or which explicitly factorize the two spaces.

We consider a situation where points in the data space may be denoted as $(x, y) \in (\mathcal{X}, \mathcal{Y})$, where $\mathcal{X}$ is continuous and $d_x$-dimensional, i.e. $x_i \in [0, 1]^{d_x}$, and $\mathcal{Y}$ is discrete and $d_y$-

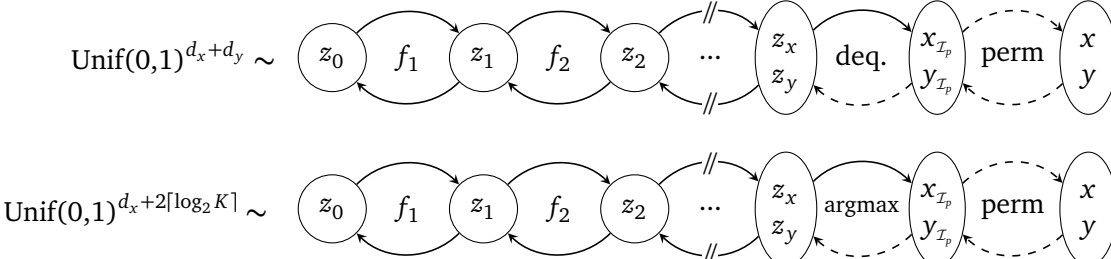

Figure 9: The variational dequantization and argmax surjection transform architectures described in sections 3.5.1 and 3.5.2. The permutation transform can either be a stochastic permutation or a sort surjection, in which case the arrow in the inverse direction would be solid.

dimensional, i.e. $y_i \in \prod_{i=1}^{d_y}\{0, ..., N_i - 1\}$.

### 3.5.1 Variational Dequantization

One option is to introduce a surjective transform that adds noise around the discrete values in the inverse direction. The result can then be appended to the other continuous features, and the normalizing flow can be used to learn correlations between the continuous and the discrete components. This process is often referred to as *variational dequantization* [53–56]. The forward and backward transforms are [1]

$$
\begin{aligned}
p_{\text{deq}}(x, y|z) &= \delta(x - z_x)\,\mathbb{I}_{F(y)}(z_y)\,, \\
q_{\text{deq}}(z|x, y) &= \delta(z_x - x)\,q_{\text{deq.}}(z_y|y)\,, \\
\mathcal{V}_{\text{deq}}(x, y, z) &= -\log q_{\text{deq.}}(z_y|y)\,,
\end{aligned}
\tag{18}
$$

where $\mathbb{I}_{F(y)}$ is the indicator function over the set

$$
F(y) = \{y + u_y | u_y \in [0, 1)^d\}
\tag{19}
$$

and the support of $q_{\text{deq.}}(z_y|y)$ is restricted to $F(y)$. The continuous latent variables are thus split up into two groups, $z_x = x \in [0, 1)^{d_x}$ and $z_y \in [0, 1)^{d_y}$ which get mapped to $y$. The resulting model is illustrated in figure 9.

Variational dequantization leads to a nonvanishing bound looseness, but a more flexible implementation of the dequantizing distribution $q_{\text{deq.}}(z_y|y)$ can reduce its size [57]. In our experiments, we include a model in which the dequantizer $q_{\text{deq.}}(z_y|y)$ samples $u_y$ uniformly, and one in which it is sampled by an auxiliary flow model conditioned on $y$. In the former case, the main flow model is tasked with learning a discontinuous probability distribution, while in the latter case the auxiliary flow can populate the disjoint sets $F(y)$ such that the distribution over $z_y$ is smoother. Both flows are trained jointly during the optimization of eq. (7).

### 3.5.2 Argmax Surjection

Variational dequantization is particularly well-suited for ordinal discrete data, where adjacent categories are often associated with similar likelihoods. However, in most situations in particle physics, the discrete data at hand are (derived from) quantum numbers which are fundamentally categorical. Dequantization may thus not be the optimal method of treating them. In [58] an *argmax surjection* was introduced to handle categorical data. This transform makes minimal assumptions about the topology of the discrete features. Individual categories are equidistant, and the continuous space is evenly partitioned.

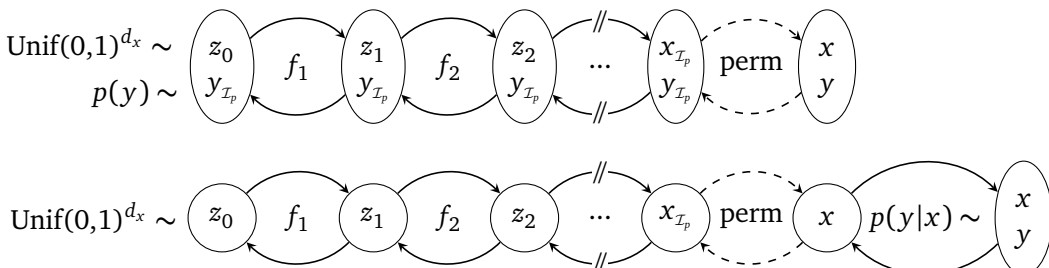

Figure 10: The factorized architectures described in section 3.5.3. The permutation transform can either be a stochastic permutation or a sort surjection, in which case the arrow in the inverse direction would be solid.

In the context of fully equidistant categories, the dimension $d_y$ of $\mathcal{Y}$ loses its meaning, and without loss of generality we can instead consider $y \in \{0, ..., K-1\}$, where $K = \prod_{i=1}^{d_y} N_i$. The transforms of the argmax surjection are also given by eq. (18), and an illustration of the architecture is shown in figure 9. However, the supporting set is now given by

$$F(y) = \left\{ z_y \middle| \operatorname*{arg\,max}_{k \in \{0, ..., K-1\}} (z_y)_k = y \right\}. \tag{20}$$

That is, $z_y \in [0,1]^K$ and the argmax operation selects $k$ if $\forall_{i \neq k} (z_y)_i < (z_y)_k$, where $(z_y)_i$ is the $i$th component of $z_y$.

A major downside of this approach is the fact that the dimensionality of the latent space is now $d_x + K$ instead of $d_x + d_y$. In the experiments we perform here (where $K = 7680$), and in most other situations, a latent space of this magnitude is not manageable. The authors of [58] propose to instead model a binary decomposition $y_B = \{0,1\}^{\log_2 K}$, such that every bit requires only two latent dimensions corresponding with the values 0 and 1. The total required latent space dimension then reduced to $d_x + 2\lceil \log_2 K \rceil$ (34 in our experiments). Note that this choice essentially represents a compromise between symmetry between the individual labels, and the dimensionality of the problem. One could thus implement other decompositions of the categorical space that would lead to a different balance.

Note that, during sampling, it is now possible for the model to generate binary encodings that correspond with categorical labels that are outside the range of the data. Of course, the model should learn to assign very small likelihoods to such events. If this happens, the complete event is rejected, and a new one is generated.

As in section 3.5.1, we perform experiments with a uniform argmax dequantizer, as well as with an auxiliary flow in an attempt to find the configuration that minimizes the bound looseness. The dequantizer is restricted to the supporting set of eq. (20) by first sampling $z_y' \in ([0,1]^2)^{\lceil \log_2 K \rceil}$. Then, a variable transform is performed to restrict the result to the set $F(y)$. For example, if $z_{y,1} > z_{y,2}$ is required to reproduce a particular bit, the transform is

$$\begin{aligned} z_{y,1} &= z_{y,1}', \\ z_{y,2} &= z_{y,1}' z_{y,2}'. \end{aligned} \tag{21}$$

The transform from $z_{y,1}', z_{y,2}' \rightarrow z_{y,1}, z_{y,2}$ induces an additional Jacobian factor $\log|J| = -\log z_{y,1}'$ that is incorporated in the evaluation of $q_{\text{deq.}}(z_y|y)$.

### 3.5.3 Factorized Models

An alternative approach is to explicitly factorize the continuous and discrete densities, leading to models of which the likelihood can be evaluated exactly. The first option is to factorize the

joint density as

$$p(x, y) = p(y)\, p(x|y). \tag{22}$$

In this case, the categorical distribution $p(y)$ is straightforwardly extracted from the data. The conditional continuous density $p(x|y)$ can be modelled by a normalizing flow that is conditioned on the category $y$.

One advantage of this method is that it is easily combined with that of section 3.4 in case the data contains variable numbers of objects that have both continuous and discrete features, i.e. particles with a four-vector and an identity. Furthermore, it is guaranteed that the marginalized categorical distribution $p(y)$ matches the training data exactly, which is not the case for the previous methods. We refer to this approach as a *mixture* model.

The second option is to instead factorize the joint density as

$$p(x, y) = p(x)\, p(y|x). \tag{23}$$

Now, $p(x)$ is the same type of normalizing flow as was considered in section 3.3, trained by ignoring the discrete features. The conditional distribution $p(y|x)$ can be implemented as a neural network that predicts the categorical probabilities of $y$ given an instance of $x$. This task is in fact identical to a multi-class classification problem, where a neural network is trained to optimize the categorical cross-entropy between real and predicted labels. We thus refer to this model as a *classifier*. Both factorized model architectures are illustrated in 10.

### 3.5.4 Experiments

We perform experiments with all methods described in the previous section, using the full training dataset. The continuous features of the four-gluino events are appended by the helicity configuration, which is encoded into a single category by interpreting it as a binary string, and by the colour ordering, which is converted into its Lehmer code [59].

For the variational dequantization models, the helicity and colour labels are kept separate. For all other models, the labels are combined into one, which is then decomposed into its binary representation for models with the argmax surjection. Following the discussion of section 3.4.1, we train the mixture model using both the likelihood and balanced prescriptions. For all cases except the classifier, we train models with both a stochastic permutation transform and a sorting surjection. The classification model is instead composed of the best-performing normalizing flow with ordering surjection trained on the full training dataset from section 3.3, and of a classifier consisting of a multilayer perceptron with 3 hidden layers of 256 nodes and ReLU activation functions, followed by a final softmax activation. It is trained following the same procedure as the normalizing flow, but with an initial learning rate of $10^{-5}$.

Figure 11 shows the test log likelihood of all models. We observe especially good performance from the mixture model with sorting surjection. A likely explanation is that the mixture models benefit from the fact that the marginalized discrete distributions are correct by construction, while all other models need to learn them implicitly. The classifier model performs slightly better than those using variational dequantization or argmax surjections, which necessarily suffer from nonvanishing bound looseness. Interestingly, among them, the dequantization options seem to be preferred even though the discrete features are of categorical nature. The most plausible explanation is that the normalizing flow is expressive enough to learn the dequantized distribution of the helicity and colour labels, while the argmax model suffers from the much larger required latent space. The choice of populating the supporting sets with a flow instead of a uniform distribution only leads to marginal improvements, which is again aligned with the observation that the baseline normalizing flow is expressive enough to learn the discontinuous distributions presented to it in the uniform case.

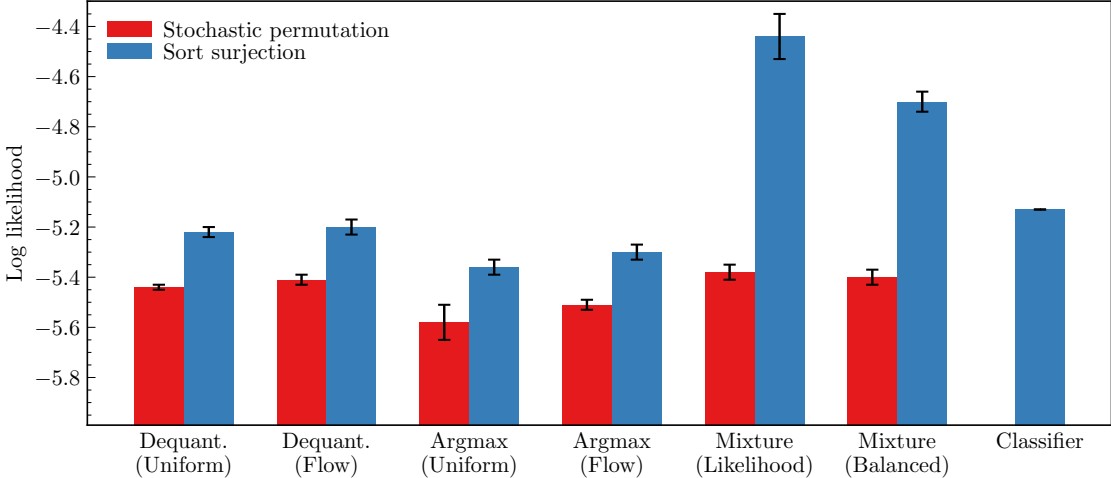

Figure 11: Test log likelihoods (higher is better) of various generative models for density estimation on mixed continuous-discrete data applied to four-gluino events, including helicity and colour labels. The bars show the average results over three independent runs, and the error bar indicates the standard deviation.

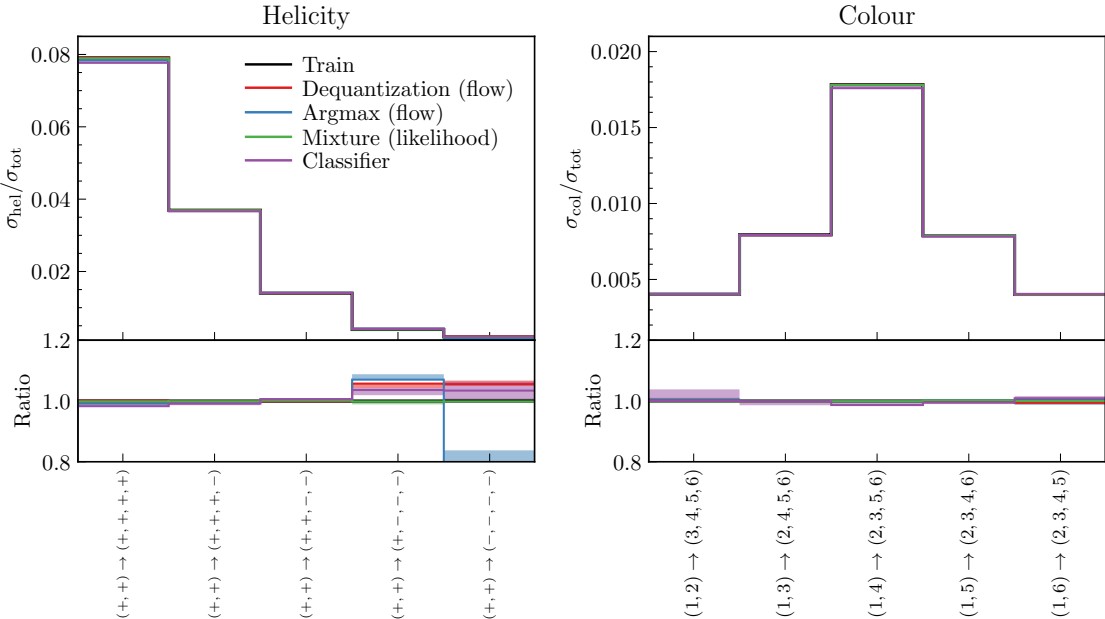

Figure 12: The marginalized discrete likelihoods of a subset of helicity and colour configurations as predicted by models with variational dequantization (red), argmax surjection (blue), the mixture model (green) and the classifier model (purple). In all cases, a sorting surjection was used. The error bands correspond with variations between three independent runs.

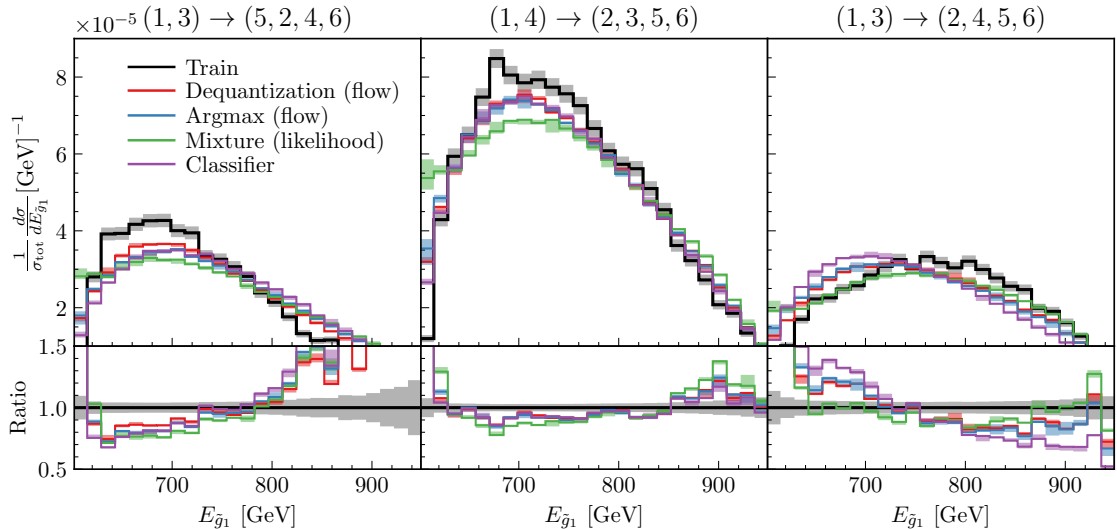

Figure 13: The gluino energy spectra of a selection of colour orderings as predicted
by the same models as in figure 12.

Figure 12 shows the marginalized discrete likelihoods of a subset of helicity and colour configurations for the best-performing models of all different types. Note that the mixture model is aligned with the training data by construction. All other models largely succeed in learning the marginalized discrete likelihoods, with differences occurring mostly in categories with low frequency.

Figures 13 shows the gluino energy spectra for a selection of colour orderings, while figure 14 shows the distribution of the spin-sensitive observable $\Delta\psi_{(12)3}$ for a selection of helicity configurations. The observable $\Delta\psi_{(12)3}$ is defined as the angular separation between the plane spanned by $p_{\tilde{g}_1}$ and $p_{\tilde{g}_2}$, and the plane spanned by $p_{\tilde{g}_1} + p_{\tilde{g}_2}$ and $p_{\tilde{g}_3}$. None of the models show substantially better or worse performance. Note that the per-category training statistics are quite limited. The performance of all models could likely be improved by the techniques explored in [11, 45, 46].

## 4  Anomaly Detection

One important application of ML-based density estimation methods is their use as model-agnostic anomaly detectors. Their objective is the identification of events that can be considered outside the Standard Model (SM) density, and may thus point to new physics beyond the SM (BSM). The use of an explicit SM density estimator has been successful in a variety of anomaly detection tasks [15, 21–23, 60–63]. Here, we apply the methods set out in the previous section to the datasets of the Dark Machines Anomaly Score Challenge [38]. In it, numerous ML-driven methods were considered to identify anomalous events from a variety of BSM models after being trained on SM background events.

Anomaly detection with an explicit likelihood estimator is accomplished by relying on the principle that a model trained on SM events should assign small likelihoods to out-of-distribution events. One thus defines an anomaly score as

$$s(x) = \frac{\log p(x) - \log p_{\min}}{\log p_{\max} - \log p_{\min}}, \tag{24}$$

where $\log p_{\max}$ and $\log p_{\min}$ are respectively the largest and smallest likelihoods assigned to the

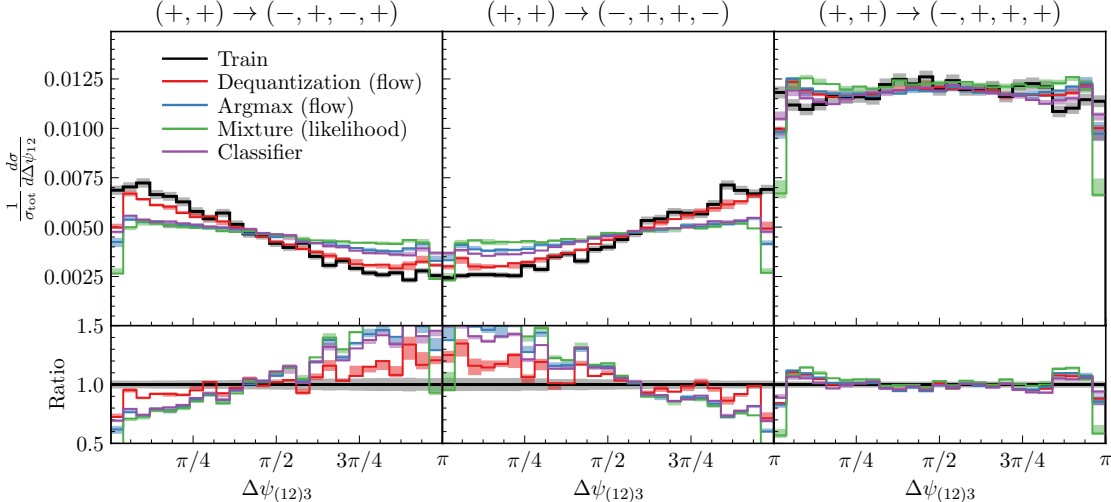

Figure 14: The distribution of the $\Delta\psi_{(12)3}$ observable defined in the text of a selection of helicity configurations as predicted by the same models as in figure 12.

event samples evaluated in the inference dataset. A cutoff point $s_{\text{cut}}$ can then be determined such that events with $s(x_i) < s_{\text{cut}}$ are classified as anomalous. To assess performance, one can then compute a background efficiency $\epsilon_B$ and a signal efficiency $\epsilon_S$ for a given value of $s_{\text{cut}}$, which represent the fraction of events maintained after application of the cut. Equivalently, the signal efficiency may be evaluated as a function of the background efficiency, $\epsilon_S(\epsilon_B)$.

## 4.1 Dataset

The Dark Machines Anomaly Score Challenge training datasets are derived from over 1 billion 13 TeV simulated SM LHC collisions. Through the application of different sets of cuts, four separate channels were identified:

- **Channel 1**: Hadronic activity with a lot of missing energy (214k events).

- **Channel 2a**: At least three identified leptons (20k events).

- **Channel 2b**: At least two identified leptons (340k events).

- **Channel 3**: Inclusive with moderate missing energy (8.5M events).

Furthermore, for testing purposes a variety of BSM signal events were generated from a variety of models, such as those containing a $Z'$ and a collection of supersymmetric models. Finally, a secret dataset is available with BSM signals unknown to the challenge participants.

The data consists of events that are each composed of a missing transverse energy $E_T^{\text{miss}}$ and its azimuthal direction $\varphi_T^{\text{miss}}$, as well as a varying number of reconstructed objects that are specified by their energy $E$, transverse momentum $p_T$, pseudorapidity $\eta$, azimuthal angle $\varphi$, and object type: jet, $b$-tagged jet, $e^+$, $e^-$, $\mu^+$, $\mu^-$ and $\gamma$.

## 4.2 Models

The data described in section 4.1 display all the properties that were incorporated in the baseline flow model in section 3. The reconstructed objects adhere to permutation invariance and can thus be either handled with a stochastic permutation or with a sort surjection as described in section 3.3. Furthermore, the number of objects is variable, which can be dealt with using

the techniques of section 3.4. Finally, the object type represents a categorical feature which can be dealt with using any of the methods of section 3.5.

We preprocess the training data by normalizing all the features to zero mean and unit standard deviation. Since the phase space is not as straightforwardly constrained as in the case of the matrix element-level case of section 3, we opt to replace the multivariate uniform base distribution of the baseline normalizing flow by a multivariate standard Gaussian. The number of objects in the feature space is restricted to a fixed $N_{\max}$, meaning that the $N_{\max}$ objects with the largest $p_T$ are included. Including $E_T^{\mathrm{miss}}$ and $\varphi_T^{\mathrm{miss}}$, the dimensionality of the baseline normalizing flow is $2 + 4N_{\max}$. The events with fewer than the maximum number of objects have a number of empty slots, which are filled with NaNs. As described in section 3.4, the baseline normalizing flow passes these NaNs without affecting them[9], and a dropout layer is included immediately after the base distribution to handle them. The hyperparameters of the baseline normalizing flow remain the same (table 1). To handle the categorical features, we consider the following three models:

- **Dequantization**: The categorical features are included using uniform dequantization as described in section 3.5.1. The object types are mapped to integer numbers in $[0, 6]$, which are dequantized into the range $[-3.5, 3.5]$. This brings the total flow dimensionality to $2 + 5N_{\max}$.

- **Mixture**: The categorical features are combined with the categorical distribution used for the dropout as described in section 3.5.3. That is, the object type is mapped to an integer in $[0, 7]$, where now 0 means the absence of an object and $[1, 7]$ are the existing object types.

- **Classifier**: The discrete features are included through a separate classifier as described in section 3.5.3 with identical architecture as the one used in section 3.5.4.

All the above models include either a stochastic permutation or a sorting surjection. We set $N_{\max} = \{8, 10\}$ for the dequantization and classifier models, but restrict the mixture model to $N_{\max} = \{6, 8\}$ due to the quick proliferation of categorical configurations as the number of objects increases.

## 4.3 Results

We evaluate the performance of the models described in section 4.2 on the four channels described in section 4.1. An often-used measure of performance is the area under the curve (AUC) of the receiver operating characteristic (ROC) curve, which shows the relationships between the background efficiency $\epsilon_B$ and the signal efficiency $\epsilon_S$. However, the AUC is dominated by the model performance at large background efficiency, while in the anomaly detection context the performance at small background efficiencies is often more relevant. Viewing the signal efficiency as a function of the background efficiency, $\epsilon_S(\epsilon_B)$, the performance metric proposed in [38] is the maximum signal improvement

$$\mathrm{Max\ SI} = \max_{\epsilon_B} \epsilon_S(\epsilon_B)/\sqrt{\epsilon_B}, \quad \text{where } \epsilon_B \in \{10^{-2}, 10^{-3}, 10^{-4}\}. \tag{25}$$

Figure 15 shows the per-channel max SI scores of the models described in section 4.2 and of the flow model used in [21], evaluated on the test signals and on the secret dataset.

We first focus on the test signals. As expected, in almost all cases the models with stochastic permutation outperform those with sort surjection in channels 1, 2a and 2b which are all

---

[9]During the evaluation of the MADE network and the classifier described in section 3.5.3, NaN features are set to zero.

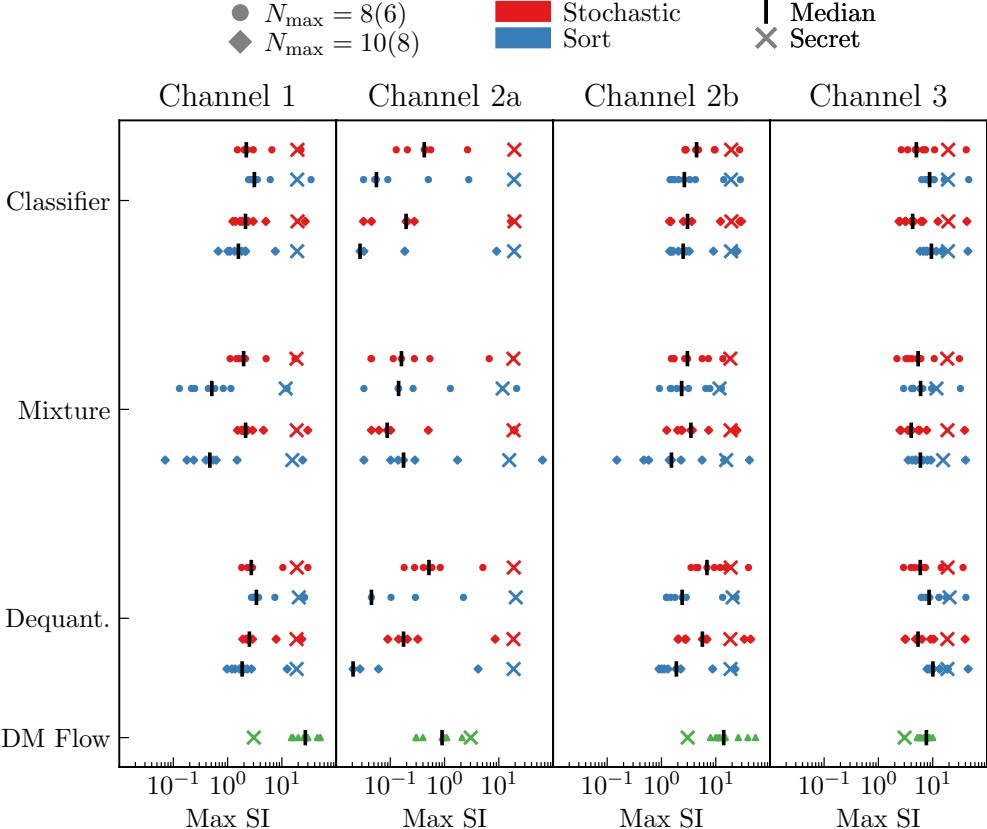

Figure 15: Max SI scores (eq. (25)) for the three models listed in section 4.2, evaluated with stochastic permutation (red) and sort surjection (blue) and varying $N_{max}$ indicated by the marker shapes. The results of the flow model used in [21] are shown in green diamonds. The black bar indicates the median performance of the BSM signals. The performance on the secret dataset is indicated with a cross.

limited by training statistics. On the other hand, channel 3 provides much more data, which benefits the sort surjection models. Similarly, models with larger $N_{max}$ tend to perform worse than their counterpart with smaller $N_{max}$ in channels 1, 2a and 2b, again due to limited training data. In channel 3 the performance is very similar, indicating that the softest objects in the event are less relevant than the hard ones for the purposes of distinguishing BSM signals from SM background. Among the treatments of the discrete features, we find that the mixture model underperforms in all channels. This is likely caused by the proliferation of possible discrete configurations which all receive their own embedding, and would thus likely require much more training data. Note that the stochastic permutation transform does not offer much help in this specific case, because it causes many new categories to appear that were not present in the sorted data.

When comparing to the flow model of [21], which was one of the best-performing models [38], we find a varied picture. The model of [21] significantly outperforms the models considered here in the low statistics channels, while sorted classifier and dequantization models outperform it in channel 3. There are several differences between the models, but the most substantial one is the treatment of differing numbers of objects. While the models considered here use the method considered in 3.4, the model in [21] used a dequantization-like procedure, where the features of missing objects are filled by out-of-distribution noise. This procedure leads to non-vanishing bound looseness, but it appears to provide better average

performance for low statistics. On the other hand, in cases where large amounts of training data are available the method of 3.4 appears to be preferred due to its access to the exact likelihood.

Shifting our attention to the secret dataset, the results differ significantly. The model of [21] underperforms on the secret dataset in channels 1, 2a and 3, while for essentially all models considered here the performance on the secret data is consistently high. In fact, in channel 3 all of them outperform the best-scoring models considered in [38], and some of them obtain the best performance in channel 2a. The secret dataset consists of a wide variety of signals including, for instance, fully unphysical events. It appears that the current models are better suited to detect such anomalies.

# 5 Conclusions

Normalizing flows have shown great promise in their application in various areas of particle physics due to their simultaneous capabilities as event generators and density estimators. However, their architecture does not provide much flexibility in terms of modelling peripheral features that are commonly associated with collision events. In this paper we explored, among other things, the addition of surjective and stochastic transforms as part of the usual normalizing flow architecture with the goal of increasing its flexibility.

In section 3 we considered the matrix element-level process $gg \to \tilde{g}\tilde{g}\tilde{g}\tilde{g}$ which displays four-fold permutation symmetry and rich discrete colour and spin spectra. We explored enforcing permutation symmetry through a stochastic permutation transform or a sort surjection, and found that both are beneficial, but the correct choice depends on the available training data. To incorporate the discrete features, we considered two surjective transforms in the form of variational dequantization and an argmax surjection, as well as two alternatives in the form of factorized models. We find that the exact likelihood evaluation offered by the factorized model leads to better performance. Finally, we also considered the issue of varying dimensionality and introduce a surjective transform with vanishing bound looseness to handle this situation.

In section 4 we applied these techniques to the objective of anomaly detection in the context of the Dark Machines Anomaly Score Challenge [38], comparing to the performance of the flow model used in [21] which was one of the best-performing models. We find results that are largely consistent with the conclusions of section 3, and achieve substantially better results on the secret dataset, outperforming all models considered in [38] in channels 2a and 3.

We believe that the techniques and the assessment of their application to typical collision events presented here will help improve future generative modelling and density estimation. While many practical applications of normalizing flows have already been explored, we expect that their general applicability will find many more use-cases in the future.

# Acknowledgements

I would like to thank Melissa van Beekveld for evaluating performance on the Dark Machines Anomaly Score Challenge secret dataset. This work was supported by the European Research Council (ERC) under the European Union's Horizon 2020 research and innovation programme (grant agreement No. 788223, PanScales).

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
