# Peer review of "Event Generation and Density Estimation with Surjective Normalizing Flows"

_SciPost Physics, doi:SciPost Phys. 13, 047 (2022)_

## Round 1 · Referee Report · Anonymous (Referee 1) · 2022-6-3

Strengths

  1. First physics application of the SurVAE framework to enhance the expressivity of both generative models and density estimators.

  2. Using stochastic layers to parametrize permutation invariants represents an interesting and novel approach that is different from the DeepSets approach.

  3. Novel and exciting approach to tackle varying dimensions in the dataset.

Weaknesses

See in the requested changes.

Report

Thanks to the author for the interesting paper. The presented approach and the results are auspicious and offer great potential for future applications for both event generation purposes and anomaly detection. For this reason, the paper is worth publishing. However, I would ask for some minor revisions/clarifications (see below).

Requested changes

Introduction:

  1. The author should cite the other applications of NF in the area of HEP: unfolding (2006.06685), calculation of loop integrals (2112.09145), bayesian flows for error estimation (2104.04543)

  2. NF citations: development of INNs (1808.04730)

Section 2:

  1. Eq.(6) is quite didactic in its presentation to give the reader an idea of where the different expressions come from. Along this line, I would suggest mentioning that the equality in the first line is due to the Bayes theorem and the fact that the integral Int dz q(z|x) is integrated out on the left-hand side (as log(p(x)) does not depend on z) and equals to 1. This would make it a more clear.

Section 3:

  1. Although it might be straightforward, the author might add 1-2 example Feynman diagrams relevant to the process in eq.(8).

  2. p.6: typo between Eq.(8) and (9): "We choose use the polar and..." -> "We choose to use the polar and..."

  3. p6: last sentence on the page: "due to the finite size of the phase space." The phase space is not finite in general. However, in your case, you remapped the phase-space onto a unit-hypercube and is thus finite. This should be clarified in the text.

  4. p7: typo: ".., of which the parameters are also listen in table 1,.." -> ".., of which the parameters are also listed in table 1,.."

  5. p8: formulation "..with varying numbers of size of the training data..." -> "..with a varying size of the training data..."

  6. Plots on p9: You nicely show error-bars in Figure 5. This should also be present as some error envelope in Figure 4, right? If so, this would be a nice add-on. Further, there is a tendency of decreasing error for an increasing amount of training data, which makes sense. But, counter-intuitively, the error bar for 500k is rather big. Do you have any idea why this is happening or is this just coincidence?

  7. p11, section 3.4: "..to stochastically drop a subset of the latent variables." Does stochastically mean that a fixed subset is dropped stochastically or that the subset itself is choosen stochastically each time? For instance, let's assume you have the latent space (z1,z2,z3,z4), where you need all variables for one process and only 2-dimensions for the other. So do you randomly decide whether to use (z1,z2,z3,z4) or the subset (z1,z2) and its always {z1,z2}, or do you also randomly pic the latent dimensions in the subset? So do you sometimes pick (z1,z4) or (z2,z3), etc. I assume you just randomly pick a fixed subset. Maybe you can clarify this in the text.

  8. p11, just right before Eq.(14) in the text: "However, if the distributions.." -> in the second "z" the "I" is not written as \mathbb like the others.

  9. p17, figure 10: In the legend, it should be "Sort surjection" and not "Soft surjection".

  • validity: high
  • significance: high
  • originality: good
  • clarity: high
  • formatting: excellent
  • grammar: excellent

Author:  Rob Verheyen  on 2022-06-17  [id 2586]

(in reply to Report 1 on 2022-06-03)

I'd like to thank the referee for their thorough reading of the paper. Below are a few comments where relevant, all other points have been incorporated in the text.

9- I agree that an error envelope is a good idea. I have added them to all plots in section 3. I do indeed believe the large error bars at 500k are a coincidence. Indeed, their size globally decreases as the training statistics increase, but the fluctuations are significant.

10- This depends on the presence of the relevant configurations in the data, and the interaction with the permutation layer. The subsets $(z_11,z_4)$ and $(z_2,z_3)$ are different dropout configurations, which each get their own dropout probability $p_{\mathcal{I}_{\downarrow}}$. If the features are permuted afterward, both of those latent space configurations could end up in the same feature space configuration. In that case, the dropout probabilities must be adjusted such that the model correctly reproduces the feature-space dropout configurations. I have added a paragraph explaining this below eq. 12

---

## Round 1 · Referee Report · Anonymous (Referee 2) · 2022-6-8

Strengths

1- The paper gives a detailed introduction to normalizing flows, variational autoencoders, and their loss functions to non-experts. Especially eq. 7 is very useful to explain the performance of different setups in later parts of the paper.

2- The paper extends the use of normalizing flows, which are bijective by construction, to cases that have a) permutation invariance ; b) varying dimensionality of features; and c) discrete features. All of these cases have applications in particle physics.

Weaknesses

1- Some descriptions of the surjective layers are hard to understand and would profit from an improved explanation

2- The section on anomaly detection does not describe the method of anomaly detection at all.

Report

The paper discusses surjective extensions of normalizing flows to particle physics. Normalizing Flows are usually bijective functions that are learned by neural networks. They have been shown to have great performance in anomaly detection, phase space integration, event generation, etc, already. However, because of their bijective nature, their application is constrained to certain cases. The paper here adds surjective pieces to the transformation and then allows the flow to handle cases with - permutation invariance - varying dimensionality of the feature space - discrete features The author shows how these building blocks improve the performance of normalizing flows on different physics datasets and gives a detailed comparison on the performance of different realizations of the respective surjections. He also explains the origin of differences. These surjections form very valuable additions to normalizing flow networks, which are especially promising to use in high-energy physics applications. I definitively think that the results should be published, however, I also think that some of the aspects should be better explained, so that they are more clear for the reader. Please find my questions together with the requested changes below.

Requested changes

1- Regarding the sorting surjection, the author says "the sort surjection orders $x$ following some predicate". It is unclear to me how this is set: Is that a fixed permutation, that is defined at the beginning? Or is that something that is defined at runtime on an event by event basis (like the jet with largest $p_T$ will be sorted to the first position etc). Please explain.

2- Section 3.3.1: Even though there is a permutation symmetry among the gluinos, there should be a ranking of the four gluinos in energy in each event. Would it make sense to train the 'no permutation' flow on samples that have been sorted in this feature? So the first 2 dimensions always belong to the most energetic gluino etc.? On a related note, I wonder how the energy spectra of the most energetic gluino, the 2nd most energetic gluino, ...etc look like when sampled from the surjective flow.

3- Section 3.3.1: Where is the "factor of 24" referring to and coming from? Please explain in more detail.

4- Section 3.4: This is one of the sections that was confusing to me. $\mathcal{I}_{\downarrow}$ is defined as "a set of dropout indices", such that together with the kept indices, $\mathcal{I}_{\uparrow}$, the full list of dimensions is recovered. Equation 12 then seems to suggest that one needs to sum over the elements of $\mathcal{I}_{\downarrow}$ and that there is a probability $p_{\mathcal{I}_{\downarrow}}$ for keeping each dimension, somehow implicitly defining $\mathcal{I}_{\downarrow}$. This, however, is wrong, as it will be clear 2 pages later, shortly before section 3.5, when $p_{two}$ and $p_{four}$ are introduced. What is actually described by equation 12 are therefore setS of indices in $\mathcal{I}_{\downarrow}$, and each set has probability of $p_{\mathcal{I}_{\downarrow}}$. Please explain this better.

5- Section 3.4: When the inverse transformation adds the dimensions back in, what values are given to these dimensions? Please explain.

6- Figure 6: Shouldn't the left side of the figure show $\mathcal{I}_{\downarrow}$, the indices that are dropped after the first step, instead of $\mathcal{I}_{\uparrow}$? Please explain or correct.

7- Section 3.5.2: I'm not sure I understand the binary decomposition. The 'usual' argmax surjection is defined by considering a vector of length $K$ with entries in $[0, 1]$. The index k of the largest entry of that vector selects the value of the discrete feature. Binary decomposition now reduces the dimensionality of the problem: instead of having $K$ additional dimensions, one uses $2\lceil \log_2 K\rceil$. argmax is then applied to pairs of 2 of these dimensions, selecting if that bit entry should become a 0 or 1. With $\lceil \log_2 K\rceil$ bits, the value $k$ of the discrete feature can be encoded. What happens if this method selects a binary value, $k'$, that does not corresponds to a class: $K < 2^{k'} < 2^{\lceil \log_2 K\rceil}$? What does the author mean by "Afterwards, for all latent space pairs in $[0, 1]^2$ that represent a bit, one of the components is multiplied by the other such that the argmax operation yields the correct value for the bit."? Please clarify.

8- Section 3.5.2: Regarding the argmax surjection: If the problem for the 'usual' approach is the rapidly growing dimensionality of the space, why is the combinatorical space defined multiplicatively? Why can't one just consider an additive combination? For example, consider discrete features of dimensionality {2, 3, 4, 2}. The method discussed here considers a vector of length $2\cdot3\cdot4\cdot2=48$ and the selected index uniquely sets which combination of the 4 features is returned. Instead, one could consider adding a vector of length $2+3+4+2=11$ and performing the argmax on the first 2, then the next 3, next 4 and last 2 dimensions, thereby also selecting a unique combination of features. A normalizing flow should be powerful enough to learn the correlations between the 4 classes, or am I missing something?

9- Section 3.5.2: Regarding the argmax surjection: How is the inverse of Eq. 19 defined? How is the discrete data mapped to the latent space vector $z_y$?

10- Figure 10: There is a typo in the legend: "Soft Surjection" $\rightarrow$ "Sort Surjection".

11- Section 4.2: In the bullet point "Mixture", shouldn't the existing object types be in $[0,5]$ instead of $[0,7]$? Please clarify.

12- Section 4.3: It is completely unclear to me how to arrive at the efficiencies $\epsilon_S$ and $\epsilon_B$ starting from a flow. Readers of this paper should not also have to read [19] in order to understand that section. Please add a paragraph or two that detail the anomaly detection algorithm of [19] that has been applied here.

  • validity: high
  • significance: top
  • originality: high
  • clarity: good
  • formatting: excellent
  • grammar: excellent

Author:  Rob Verheyen  on 2022-06-17  [id 2587]

(in reply to Report 2 on 2022-06-08)

I'd like to thank the referee for their thorough reading of the paper. Below are a few comments where relevant, all other points have been incorporated in the text.

  1. It is indeed defined at runtime, on an event-by-event basis. I think this is implied by eq.11, but I have clarified the action of the sort surjection in 3.3.1 in the context of the four-gluino example, where the gluinos are sorted according to their polar angle (this information was missing).

  2. Because the sort surjection is placed at the end of the flow, this suggestion is essentially what is done in the case of the sort surjection, except that there is still a stochastic permutation in the forward direction. In essence, the sort surjection corresponds with a pre-processing of the data.

  3. The factor of 24 = 4!, the improvement factor due to stochastic permutation. I have clarified this.

  4. I am not sure I understand the confusion here. The assessment that $I_{\downarrow}$ implicitly defines $I_{\uparrow}$ is correct, and I think this is explained above eq. 14. The probabilities $p_{\text{two}}$ and $p_{\text{four}}$ were just meant to represent the probabilities to sample a 2-gluino event or a 4-gluino event, i.e. the probabilities to drop 2 gluinos or to drop no gluinos. However, this was not well explained in the text, so I have tried to clarify.

  5. The point of the latter part of 3.4 is to show that it is advantageous to place the dropout surjection right after the base distribution. It leads to tractable likelihood evaluation, basically because the base distribution is so simple. What this also means is that it does not really matter what values are assigned to the dropped indices in the inverse transform. These values would then only be used to evaluate the base distribution, but because all latent features of the base distribution are independent, the contribution to the likelihood will integrate to unity either way. In practice, the log-likelihood contribution of the base distribution with independent latent variables is $\log p(z) = \sum_i \log p(z_i)$, and dropped features just contribute with $\log p(z_i) = 0$. I expanded the end of 3.4 a little to clarify this.

  6. Correct, fixed.

  7. For the first question, it is indeed possible that the flow samples discrete values that is too large to correspond with an existing categorical label. In this case, the full event should be rejected and a new one should be sampled. This should have been explained in the text, and I have added it now. For the second question, I will return to this in 9.

  8. This assessment is completely correct. However, it is important to realise that, when the discrete labels are categorical, the dimensionality of the discrete space becomes meaningless. In the given example, the space {2, 3, 4, 2} corresponds with 48 labels that are all equally different. In particle physics, a natural dimensionality is often given by i.e. the features of individual particles (for example, the particle id of every particle in the event). However, sometimes this is not quite so straightforward (i.e. colour flows). In general though, a discrete label of {jet, jet, electron, photon} contains the same information as {2jets+electron+photon}, and it is not obvious which representation is easier to train a NF on. The 'usual' approach would thus correspond to mapping these features to a space {48}, while the binary approach would map the features to {2,2,2,2,2,2}. The dequantization approach instead retains the original mapping. Other encodings are thus also possible, they basically correspond to a trade-off between symmetry and dimensionality. I have added a note about this to the text.

  9. This is explained in the last paragraph of 3.5.2. The procedure is similar to the case of variational dequantization, where the dequantizer needs to fill the supporting set $F(y)$. In this case, that is accomplished by letting it fill the full hypercube, and then performing a transformation that restricts to $F(y)$. I had tried to explain this in words, but it was not clear enough, so I have added a more explicit example. I hope that this helps to clarify.

  10. Correct, fixed.

  11. There was indeed a typo here (it said [0,6]), but the the objects types should be [0,7], because objects can either be empty (0) or have one of seven types [1,7].

  12. This is a fair point, I have added a paragraph that explains the process above section 4.1.

---

## Round 2 · Referee Report · Anonymous (Referee 1) · 2022-6-23

Report

The author has incorporated and answered all my requests satisfactorily. Therefore, I recommend the paper for publication

---

## Round 2 · Referee Report · Anonymous (Referee 2) · 2022-7-1

Report

The author has addressed all the points that I raised to my satisfaction. The explanations in the manuscript improved and are more clear now. I can recommend this paper for publication.

---

## Round 2 · List of Changes

All points raised by the referees have been addressed.

---

## Editorial Decision

published